# Persicae Semen Promotes Bone Union in Rat Fractures by Stimulating Osteoblastogenesis through BMP-2 and Wnt Signaling

**DOI:** 10.3390/ijms24087388

**Published:** 2023-04-17

**Authors:** Jae-Yun Jun, Jae-Hyun Kim, Minsun Kim, Sooyeon Hong, Myunghyun Kim, Gwang-Hyun Ryu, Jae Ho Park, Hyuk-Sang Jung, Youngjoo Sohn

**Affiliations:** 1Department of Anatomy, College of Korean Medicine, Seoul 02447, Republic of Korea; 2Department of Pharmaceutical Science, Jungwon University, Goesan-eup 28024, Republic of Korea

**Keywords:** Persicae semen, femoral fractured rat, MC3T3-E1, BMP-2, Wnt, RUNX2

## Abstract

Fractures cause extreme pain to patients and impair movement, thereby significantly reducing their quality of life. However, in fracture patients, movement of the fracture site is restricted through application of a cast, and they are reliant on conservative treatment through calcium intake. Persicae semen (PS) is the dried mature seeds of *Prunus persica* (L.) Batsch, and in this study the effects of PS on osteoblast differentiation and bone union promotion were investigated. The osteoblast-differentiation-promoting effect of PS was investigated through alizarin red S and Von Kossa staining, and the regulatory role of PS on BMP-2 (*Bmp2*) and Wnt (*Wnt10b*) signaling, representing a key mechanism, was demonstrated at the protein and mRNA levels. In addition, the bone-union-promoting effect of PS was investigated in rats with fractured femurs. The results of the cell experiments showed that PS promotes mineralization and upregulates RUNX2 through BMP-2 and Wnt signaling. PS induced the expression of various osteoblast genes, including *Alpl*, *Bglap*, and *Ibsp*. The results of animal experiments show that the PS group had improved bone union and upregulated expression of osteogenic genes. Overall, the results of this study suggest that PS can promote fracture recovery by upregulating osteoblast differentiation and bone formation, and thus can be considered a new therapeutic alternative for fracture patients.

## 1. Introduction

Bones are important organs that provide support for the body, protect internal organs, and enable body movement [1]. Recently, with the development of medicine and industry, the number of osteoporotic patients is rapidly increasing due to an increase in size of the elderly population [2]. Because osteoporosis is a direct cause of fractures, the number of fracture patients is also increasing, but most fracture patients restrict the movement of their fracture site with a cast, relying solely on conservative treatments through calcium intake for recovery [3]. Fractures cause extreme pain to a patient and impair movement, thereby significantly reducing their quality of life, and cases of severe traumatic fracture may be accompanied by organ damage and shock [4]. Furthermore, not only do direct medical expenses associated with fractures add to the economic burden of households, but limitations in daily activities resulting from fractures can also cause financial strain by reducing individuals’ economic productivity [5]. Parathyroid hormone (PTH) is sometimes prescribed for the treatment of fractures of high severity, but this is accompanied by serious side effects such as muscle weakness and vomiting, and the cost is substantial, so it is not suitable for long-term administration [6]. For these reasons, the development of a therapeutic agent to promote bone union that is based on natural products is a major area of research.

Osteoblasts are derived from mesenchymal stem cells and play a major role in fracture repair. Bone morphogenetic protein-2 (BMP-2) is known to be a major mechanism of osteoblast activation and bone formation [7]. BMP-2 activity leads to the phosphorylation of Smad, allowing its translocation to the nucleus, and activates p38, upregulating Runt-related transcription factor 2 (RUNX2) [8], which is recognized as a master transcription factor for bone formation [9]. Another growth factor family that promotes osteoblast differentiation of mesenchymal stem cells is Wnt/β-catenin signaling [10]. β-Catenin, after accumulating in the cytoplasm to a particular threshold, is then translocated into the nucleus to activate T-cell factor/lymphoid enhancer factor (TCF/LEF), consequently inducing the upregulation of several bone-related factors, including RUNX2. Thereafter, RUNX2 induces the expression of osteoblast-related genes, such as alkaline phosphatase (*Alpl*), osteocalcin (*Bglap*), and bone sialoprotein (*Ibsp*) [11].

Persicae semen (PS) is the dried mature seeds of *Prunus persica* (L.) Batsch and *P. davidiana* (Carr.) Franch (also known as peach fruit), belonging to the Rosaceae family. PS is traditionally used as a treatment to accelerate blood flow; it has also been used for treating conditions such as neuralgia, muscle pain, menstrual pain, and constipation [12]. In recent studies, PS demonstrated pharmacological potential based on anti-tumor [13] and anti-thrombotic [14] effects, but effects on osteoblast activation and fracture recovery have not yet been shown.

In this study, we investigated the impact of PS on osteoblast activation, with a focus on the BMP-2/Wnt/RUNX2 pathway, which plays a crucial role in this process. In addition, the bone-union-promoting effect and in vivo mechanism were analyzed by inducing a femoral-fractured rat model and administering PS. Finally, we demonstrated that PS promotes osteoblast activity, promoting bone union, and can thus represent a new alternative for fracture treatment in the future.

## 2. Results

### 2.1. PS Enhances the Formation of Mineralized Nodules in MC3T3-E1 Cells

The effects of PS on MC3T3-E1 cell viability were assessed using the cell counting kit-8 (CCK-8) assay, and the results show that PS did not have any effect on cells following exposure for 1, 3 or 7 days (Figure 1A–C). To determine the effect of PS on mineralization, cells were incubated in medium containing PS for 14, 18, and 21 days, and deposited calcified nodules were then stained with alizarin red S solution (Figure 1D). On day 14, 100 μg/mL of PS promoted the formation of calcified nodules faster than the osteogenic medium alone. On day 18, cells treated with 25–100 μg/mL of PS showed relatively high calcium deposition compared to cells treated with osteogenic medium. In addition, on the 21st day, cells treated with 50–100 μg/mL of PS filled the plate to form calcified nodules, but microscopic deposition was only observed for the cells treated with osteogenic medium. After extracting data for alizarin red S dye staining in each experiment, we found that cells treated with PS showed increased staining intensity compared to those treated with osteogenic medium alone on days 14, 18, and 21. Another staining method for mineralized nodules, Von Kossa staining, was also used to investigate the osteoblast differentiation-promoting effect of PS (Figure 1F). After measuring the stained area, which was the same as for alizarin red S, we concluded that PS significantly increases the number of mineralized nodules in a concentration-dependent manner (Figure 1E,G).

### 2.2. PS Increases the Expression of BMP-2 Signaling in MC3T3-E1 Cells

BMP-2 is a key transcription factor known to induce RUNX2 activity during osteoblast differentiation. Thus, to examine the effect of PS on the BMP-2 signaling pathway, we performed Western blot. The levels of BMP-2, RUNX2, and Osterix proteins were increased through PS treatment and significantly increased, particularly for PS at 100 μg/mL. Phospho (p)-Smad showed a tendency to increase in response to PS, but the difference was not significant. On the other hand, the phosphorylation of p38 was significantly increased by PS treatment. (Figure 2A,B).

### 2.3. PS Increases the Expression of Wnt/β-Catenin Signaling and Osteoblast-Related Genes in MC3T3-E1 Cells

Wnt/β-catenin signaling is another mechanism that plays a major role in osteoblast differentiation. We verified how PS affects the expression of Wnt/β-catenin-related indicators via reverse transcription–polymerase chain reaction (RT-PCR). PS upregulated the expression of *Wnt10b*, *Ctnnb1*, *Dvl2*, and *Lrp*6 (Figure 3A). After quantifying the expression of each mRNA through normalization with *Actb*, significant increases were shown following treatment with PS at 100 μg/mL (Figure 3B). Activated RUNX2 upregulated the expression of osteoblast-related genes such as *Alpl*, *Bglap*, *Ibsp* and *Col1al*. Compared with non-treated cells, PS treatment had an enhancing effect on all markers that was concentration-dependent. In particular, 100 μg/mL of PS showed significant effects on *Alpl*, *Bglap* and *Ibsp* (Figure 3C,D).

### 2.4. PS Promotes Bone Formation in Fracture-Induced Models

Fracture healing proceeds via three stages (inflammatory, reconstruction, and remodeling) that each occur in succession [15]. To determine the extent of recovery during drug administration, it is crucial to establish a suitable endpoint for the experiment after the fracture. If sacrifice occurs too early, callus formation will be incomplete due to inflammatory stages, and if it occurs too late, all the fractures in the control group will have undergone recovery. Therefore, setting an appropriate termination time is of the utmost importance. Before proceeding with this study, we conducted a preliminary study wherein we compared the degrees of recovery following sacrifice at 2 and 4 weeks after fracture of the femur (Figure 4A). At 2 weeks after fracture, only soft callus formation was observed, and there was no sign of bone union. On the other hand, a clear callus line was observed at 4 weeks, and, at the same time, it could be observed that the femur bone union was progressing (Figure 4B). Therefore, we decided on an experimental period of 4 weeks. To evaluate the effect of PS on promoting fracture healing, following the induction of femur fractures in rats, the oral administration of PS was carried out, and the recovery capacity was evaluated using micro-computed tomography (CT) after 4 weeks. The PS-low (L) group was administered a concentration of 50 mg/kg of PS, and the PS-high (H) group was administered 100 mg/kg of PS, while the control group received the same volume of distilled water. The micro-CT results revealed a clear fracture line and bone union compared with the control group (Figure 4C). When comparing the femurs in terms of changes in gross anatomical appearance, the PS-treated group showed faster fracture recovery than the control group, in addition to the disappearance of the callus (Figure 4D). Based on measurements of the fracture healing score through micro-CT and the gross anatomy results, the PS-administered group showed a more improved fracture recovery capacity than the control group, an effect that is concentration-dependent (Figure 4E). Following analysis to verify the change in the callus microstructure at the fracture site, the PS-L and -H groups showed significantly increased bone volume (BV) compared with the control group. In addition, the PS-H group showed significantly increased trabecular thickness (Tb.Th) (Figure 4F).

### 2.5. PS Promotes Osseous Callus Formation and Increases BMP-2, RUNX2, and OCN Expression in Femoral Tissue

To determine morphological changes in callus formation, we performed hematoxylin and eosin (H&E) staining. In the control group, the areas corresponding to the fibrous and osseous calluses were maintained at the fracture site, but in the PS-L group the area corresponding to the fibrous callus was reduced. In particular, in the PS-H group, all fibrous calluses were converted to osseous calluses (Figure 5A). Thereafter, the area of callus formation was verified through safranin O staining. The red callus was still present in the femur of the control group, but there was no safranin O-positive area in the PS-L and -H groups (Figure 5B). The expression of BMP-2, RUNX2, and OCN in femoral tissue was detected by IHC staining. In the control group, a very small positive area (brown area) was observed for each indicator. On the other hand, the PS-H group showed significant increases in all the indicators (BMP-2, RUNX2, and OCN) compared with the control group (Figure 5C–E).

### 2.6. PS Increases the Expression of BMP-2 and Osteoblast-Related Markers in Whole Blood and Serum

To investigate the effect of PS administration on bone formation markers in whole blood, we verified the expression of *Bmp2*, *Bglap*, and *Ibsp* using RT-PCR, with the result shown in Figure 6A. The PS-H group had significantly increased BMP-2 and OCN expression and slightly (not significant) increased BSP expression. To investigate the effect of PS on the expressions of osteogenic factors in serum, we measured the expression of ALP, osteonectin (OSN), and OCN using enzyme-linked immunosorbent assay (ELISA). The experimental results showed significantly upregulated ALP, OSN, and OCN expression in the PS-H group compared with the control group (Figure 6B). To verify the in vivo toxicity of PS, the expressions of aspartate transaminase (AST) and alanine transaminase (ALT) in the serum were examined. According to the experimental results, PS-L and -H did not show significant differences compared with the control (Figure 6C). The administration of PS did not significantly affect the weights of the rats every week until the end of the experiment, and there was also no significant difference in the weight of the liver (Figure 6D).

### 2.7. Standardization of PS and Determination of the Effect of Amygdalin on Osteoblasts

Obtaining scientific data on the quality control of natural materials and the reproducibility and safety of medicinal effects is very important for natural medicines. Therefore, we compared high-performance liquid chromatography (HPLC) peak data of PS and amygdalin, a well-known component of PS [16]. In the HPLC analysis results, peaks corresponding to the same retention time were detected for PS and amygdalin (Figure 7A,B). Alizarin red S staining was performed to assess whether the osteoblast differentiation-promoting effect of PS was mediated by amygdalin (Figure 7C), but the results indicated that amygdalin had no effect on bone formation (Figure 7D). Therefore, the effect of amygdalin on the osteoblast differentiation of MC3T3-E1 cells was investigated, but no significant effect on osteoblast differentiation was found, and the corresponding concentration did not show toxicity (Figure 7D,E). Amygdalin is contained in the seeds of nuclear fruits such as peaches and plums, and when it is combined with water in the process of decomposition after ingestion, the toxin HCN is generated, so it should be consumed with caution. In total, 100 mg amygdalin contains 6 mg of HCN, and the lethal dose of HCN in humans is 0.5 mg/kg. The content of amygdalin in 1 mg PS was found to be 6.8 μg. Therefore, since the concentrations of amygdalin in the concentrations of 50 and 100 mg/kg administered in this study were very low, at 0.3 and 0.6 mg/kg (HCN content; 20.4 and 40.8 μg/kg), the toxicity of amygdalin was not expected to have any significant practical impact. Consistent with these findings, it has been reported that PS administration does not have any effects on serum ALT/AST and liver weight [17].

## 3. Discussion

The purpose of this study was to characterize the effect of PS on the promotion of bone union in an animal model of fracture and in the mechanism of osteoblast differentiation. In brief, the results regarding safety indicate that PS did not show cytotoxicity toward MC3T3-E1 cells, and it did not affect the values of hepatotoxicity indices in serum. In addition, treatment with PS was found to enhance BMP-2/Smad signaling, which plays an important role in the mechanism of differentiation of MC3T3-E1 cells, which are osteoblast progenitor cells, and in bone fracture animal models. As a result of PS treatment, the expression of various osteogenic markers, such as ALP, OCN, OSN, and BSP, was upregulated. Consistent with these results, improved bone union and microstructure of the fracture site were observed, as analyzed through micro-CT and histological observations in the rat femoral fracture model.

In Korean medicine, it is considered that the main purpose of fracture healing methods is to facilitate blood flow to the fracture site. Medications are administered differently depending on the stage of fracture recovery. In the early stage of fracture, “Hwal-hyeol-yak”, a natural product that speeds up blood circulation, is administered for the healing of inflammation and rapid callus formation at the fracture site. Afterwards, “Bohyeol-yak”, which plays a role in replenishing blood, is administered to promote bone union by supplying the fracture site with copious blood [18]. Various recent studies have demonstrated that abundant and rapid angiogenesis plays a major role in fracture recovery. Angiogenesis supplies nutrients and facilitates the formation of mesenchymal cells, which in turn promotes fracture recovery [19]. PS is a representative “Hwal-hyeol-yak”; it has traditionally been used to treat lumps or swelling by accelerating blood circulation. These traditional methods are thought to act a manner that is in accordance with the philosophy of traditional fracture treatment in Korean medicine, so we expected that drugs that accelerate blood flow (such as PS) would be effective in fracture recovery.

MC3T3-E1 is an osteoblast precursor cell line extracted from Calvaria in mice. It has been used as an experimental model to validate osteoblast differentiation and bone formation in various studies [20]. MC3T3-E1 cells have various subclones, of which the differentiation of subclone 4 into osteoblasts can be induced by ascorbic acid and inorganic phosphate, resulting in the formation of a mineralized extracellular matrix about 10 days after stimulant treatment, which is an important marker for osteoblast differentiation. In this study, MC3T3-E1 subclone 4 was used, and the formation of a mineralized extracellular matrix was observed after stimulation, as in a previous study [21]. Alizarin is a dye that specifically adsorbs calcium, and since it is stained on the mineralized cell matrix, the degree of staining is proportional to the amount of calcification. In this study, PS promoted the formation of the mineralized extracellular matrix, which means that it exerted its effectiveness in the most important process of post-fracture bone union, namely osteogenesis.

BMP-2, a member of the transforming growth factor (TGF)-β superfamily, is a factor that plays a pivotal role in osteoblast differentiation and the formation of the mineralized extracellular matrix [22]. BMP-2 knockout is embryonically lethal, making in-depth studies impossible. However, recent experiments on transgenic mice in which BMP-2 is inactivated have revealed that these mice developed spontaneous, non-healing fractures and fracture recovery. This suggests that BMP-2 plays an important role in fracture recovery [23]. BMP-2 signaling is linked to both type I and type II BMP receptors. When BMP-2 binds to these receptors, it triggers the formation of a complex between the two receptors, which leads to transphosphorylation. This activation of the receptors subsequently activates mechanisms for osteoblast differentiation and bone formation through the use of Smads or MAPKs [24]. It then induces the expression of RUNX2 (also known as Cbfa1/Pebp2aA/AML3), which is known as a major factor in bone formation. RUNX2, one of the members of the Runt family of transcription factors, regulates the expression of major osteogenic factors during the transformation from mesenchymal stem cells to osteoblasts [25]. In a previous study, it was demonstrated that osteoblast maturation and ossification are impossible in mice deficient in RUNX2, and it was also demonstrated that when the expression of RUNX2 was controlled, the maintenance of bone mass in old age was impaired [26]. In this study, it was observed that PS upregulated the expression of BMP-2 and RUNX2, but only induced a slight increase in the expression of SMAD1/5, which was not statistically significant. Thus, it was concluded that PS upregulated the expression of RUNX2 through other pathways than Smad. The expression of p38 is crucial for both osteogenesis and BMP-2 expression [8]. Previous studies have shown that inhibiting p38 in osteoblasts completely prevents the gene expression and mineralization of BMP-2 and osteocalcin. Additionally, the p38 inhibitor SB203580 impairs osteoblast differentiation and the expression of osteoblast markers such as ALP and OCN in MC3T3-E1 cells [27]. This phenotype is similar to that of RUNX2 mutants, suggesting that p38 is essential for RUNX2’s function. Osterix is an osteoblast-specific zinc-finger-containing transcription factor that is known to play a major role in the differentiation of lineages into osteoblasts and chondrocytes [28]. In the embryos of mice deficient in osterix, chondrogenesis was normal, but no bone formation was observed at all, and no Col1, BSP, OSN, or osteopontin were detected [29]. In this study, PS significantly increased the expression of BMP-2, RUNX2, osterix and phosphorylation of p38 in MC3T3-E1 cells. This suggests that the mineralized extracellular matrix-promoting effect of PS is mediated by the activity of p38 and RUNX2 through the binding of BMP-2 and the BMP receptor.

The importance of the Wnt mechanism in bone formation has already been demonstrated through several studies [30]. Subsequent genome-wide association studies have demonstrated that Wnt has a direct effect on BMD, and it was recently shown that mutations in Wnt1 are involved in osteoporosis and hyperostosis [31]. In addition, missense mutations in Wnt16 were found to be involved in hip fracture and bone mass. Mice deficient in *Lrp6* are characterized by osteopenia due to their inability to activate osteoblasts, and the expression of the mutant *Lrp6* allele increases bone mass in mice [32]. β-Catenin is essential for bone formation, and it regulates both osteoblasts and osteoclasts. Although the direct target gene of β-catenin has not yet been fully verified, it is known that, together with Tcf1, it directly regulates the expression of RUNX2 [33]. In this study, PS significantly increased the expressions of *Wnt10b*, *Ctnnb1*, *LRP6*, and *Dvl2* in MC3T3-E1 cells. These results suggest that the mineralized extracellular matrix-promoting effect of PS is due to the upregulation of RUNX2 through stimulation of the Wnt-β-catenin mechanism, along with the BMP-2-p38 mechanism.

RUNX2, when activated, controls the expression of various osteogenic genes such as *Alpl*, *Ibsp*, *Bglap,* OSN (*Sparc*), and *Col1a1*. Osteoblast differentiation is closely related to the expression of bone matrix proteins [34]. Of the various bone matrix proteins, ALP and collagen are expressed in the early stage of differentiation, while BSP and OCN are expressed only in mature osteoblasts. ALP is present in almost all tissues and, in particular, the activity of ALP present in bone tissue is increased during active bone growth and is thus used as a biomarker indicative of osteoblast activity [35]. OCN, also called bone Gla protein, is currently the most useful clinical marker of bone formation. OCN is formed in osteoblasts and then deposited in the bone matrix, and part of the newly formed bone is released into the blood, indicating the degree of formation [36]. BSP is known as a representative marker gene for bone calcification and differentiation that can regulate mineral crystal formation. BSP accounts for 15% of the total protein in bone, and it is known to be involved in bone remodeling [37]. The process of bone formation includes the synthesis of type I collagen, intracellular and extracellular secretion, fiber formation, the maturation of the collagen matrix, and hydroxyapatite formation. Type I collagen synthesized in osteoblasts accounts for 85–90% of the total bone protein [38]. In addition, collagen accounts for most of the organic matter of bone tissue, and it is known that, in addition to not being able to calcify, bone tissue lacking collagen also has decreased ALP activity OCN expression. OSN is known to play an important role in the formation of extracellular matrix mineralization by strongly binding to type I collagen [39]. In addition, it is known that the formation of osteoblasts is disrupted in mutants of OSN [40]. After analyzing the effects on each osteogenic factor, we determined that PS significantly increased the gene expression of *Alpl*, *Ibsp* and *Bglap,* and increased the expression of *Col1a1*. These results indicate that PS increases BMP-2 and RUNX2 and promotes the expression of osteogenic factors, which are subfactors thereof (Figure 8).

When recovery due to treatment proceeds from the fracture line, the bone structure of the wound site is organized through the activity of osteoclasts, with the activation of osteoblasts. During this process, the observation of the tissue through micro-CT reveals that the cut bone surface becomes unclear or appears as a formed shape [41]. In addition, the volume of the formed callus decreases as bone union progresses, and the fibrous callus is converted into an osseous callus [42]. In our study, the administration of PS was shown to promote bone union, with the observation of a gross anatomical feature that disappeared after callus formation in the fractured femur. In addition, through histological analysis, it was confirmed that the callus in the femur tissue rapidly changed from an osseous callus to a fibrous callus, and the area of the callus was also reduced. In addition to the histomorphological analysis, the expression of osteogenic genes in tissues was also subjected to histochemical analysis. The administration of PS induces the expression of BMP-2, RUNX2, and OCN in the fractured femur tissue, in contrast to the control. Consistent with the histological analysis results and cellular-level studies, the expressions of BMP-2 and OCN in whole blood were significantly increased in the PS-H group compared with the fracture-inducing group. In addition, the PS-H group showed significantly increased expressions of ALP, OSN, and OCN in serum compared with the control group. In summary, these results indicate that PS promotes bone formation through the upregulation of BMP-2 signaling, resulting in effective bone union in a rat femoral fracture model. This indicates the possibility of using PS as an alternative fracture treatment in the future.

The limitations of this study are as follows: (i) In this study, the experiment was conducted up to 4 weeks after fracture induction in both the control group and the PS-administered group, and PS showed a significant bone union effect. However, given that some fractures, such as penetrating fractures and comminuted fractures, showed slower recovery due to their different characteristics, it is necessary to investigate the effects of promoting bone union through PS administration for at least 4 weeks or longer in other fracture models in the future [43]. (ii) In addition, the osteoblast-promoting effect of PS is expected to be effectively applied to bone metabolic diseases, including osteoporosis. It will be considered very important in future to verify the effectiveness of PS in menopausal and senile osteoporosis (primary osteoporosis) models as well as models of steroid-induced osteoporosis (secondary osteoporosis) [44]. (iii) In this study, we confirmed the osteoblast-activating effect of amygdalin, a major component of PS. However, this particular component alone did not produce a healing phenotype. PS is composed of cyanides, triterpenes, steroids, phenolic acids, and fatty acids, and representative components include amygdalin, prunasin, β-sitosterol, campesterol, chlorogenic acid, oleic acid, and linoleic acid [16,45,46]; among them, β-sitosterol [47], chlorogenic acid [48], and linoleic acid [49] are known to promote osteoblast differentiation. β-sitosterol upregulates RUNX2 through the induction of ERK and p-38, chlorogenic acid upregulates BMP-2/RUNX2, and finally linoleic acid upregulates RUNX2 through the Wnt mechanism to promote osteoblast differentiation. Taken together, these results suggest that the main mechanism of the osteoblast differentiation-promoting effect of PS is the upregulation of RUNX2 through various mechanisms of its constituents, and among them the pharmacological effect of β-sitosterol was found to be similar to that of PS. If the contents of β-sitosterol, chlorogenic acid and linoleic acid in PS can be precisely analyzed and extracted in future research, it may be possible to identify the specific healing phenotype of PS. (iv) PS has demonstrated the ability to effectively restore fractures; however, it remains unclear whether its efficacy is attributed to the traditional method of use, which is believed to enhance blood circulation. Therefore, further research is warranted. To clarify the fracture repair mechanism of PS based on traditional usage, it is recommended to investigate its impact on vascular endothelial growth factor (VEGF) and platelet-derived growth factor (PDGF) [50], which are angiogenic factors associated with fracture repair. Such investigations would allow us to determine the specific effects of PS on these factors and elucidate its potential role in fracture repair.

## 4. Materials and Methods

### 4.1. Reagents

Minimum essential medium-α (α-MEM) and penicillin/streptomycin (P/S) were obtained from Gibco (Gaithersburg, MD, USA). The CCK-8 assay was obtained from Dojindo Molecular Technologies, Inc. (Kumamoto, Japan). Fetal bovine serum (FBS) was supplied by Atlas Biologicals (Fort Collins, CO, USA). Bicinchoninic acid (BCA) protein assay kit, β-glycerophosphate, ascorbic acid, dimethyl sulfoxide (DMSO) and amygdalin were obtained from Sigma Aldrich (St. Louis, MO, USA). Dulbecco’s phosphate-buffered saline was supplied by Welgene, Inc. (Daejeon, Republic of Korea). Anti-RUNX2, anti-BMP-2 and anti-SMAD1/5/9 antibodies were supplied by Abcam (Cambridge, UK). Anti-t-p38 and anti-p-p38 antibodies were purchased from Cell Signaling Technology, Inc. (Danvers, MA, USA). Secondary antibodies were supplied by Jackson ImmunoResearch Laboratories, Inc. (West Grove, PA, USA). PCR primers were supplied by Genotech (Daejeon, Republic of Korea). The SuperScript II Reverse transcription kit and SYBR-Green solution were purchased from Invitrogen (Carlsbad, CA, USA). Taq polymerase was purchased from Kapa Biosystems (Woburn, MA, USA). The OCN (cat. no: LS-F12230) and the OSN (cat. no: LS-F27540) ELISA kit was obtained from LSbio (Seattle, WA, USA).

### 4.2. Preparation of PS

PS was obtained from Omni Herb Inc. (cat. no: 061900450, Seoul, Republic of Korea). Ripe seeds of peaches collected in Hebei, China, in June–July were used. The specimens of PS used in this study are stored in the natural product storage of the department of anatomy’s herbarium (No. KHU-ANA-Et009). The samples (300 g) were extracted with 80% EtOH for 3 weeks, and the extract was filtered with filter paper (no. 3; Whatman plc; GE Healthcare, Chicago, IL, USA). After this, the ethanol extract was evaporated using a rotary vacuum evaporator (EYELA, Tokyo, Japan). The extract was dried in a freeze-dryer, which yielded the dried powder (yield ratio: 2.79%). The dried powder was stored at −20 °C. PS was dissolved in DMSO, and the dose of DMSO did not exceed 1% in the cell medium.

### 4.3. Cell Culture and Cell Viability

MC3T3-E1 cells were obtained from the American Type Culture Collection (ATCC, Manassas, VA, USA). The cells were cultured in α-MEM (without ascorbic acid) supplemented with 10% FBS and 1% P/S. Cells were cultured in a humidified CO_2_ incubator at 37 °C. To assess the effect of PS and amygdalin on cell viability, 1 × 10^4^ cells in 100 μL medium per well were seeded in a 96-well plate and incubated at 37 °C for 1 and 3 days, then treated with PS (25, 50, and 100 μg/mL). After this, the cells were added to a CCK-8 assay at 10 μL per well and incubated for 2 h at 37 °C in a CO_2_ incubator. They were then measured at 490 nm using an ELISA reader (VersaMax microplate reader; Molecular Devices, LLC, San Jose, CA, USA).

### 4.4. Alizarin Red S and Von Kossa Staining

In total, 5 × 10^5^ cells in 3 mL medium per well were seeded in a 6-well plate and incubated at 37 °C for 24 h. After, the cells were treated with osteogenic media consisting of 25 μg/mL ascorbic acid and 10 mM β-glycerophosphate with or without PS (25, 50, and 100 μg/mL) and amygdalin, and replacements with fresh osteogenic media were conducted every 3 days. Calcified nodules were observed from the 14th day of culture and continuously increased until the 21st day, and the plate was stained on the 14th, 18th, and 21st days, respectively. Briefly, the staining method was as follows: The cells were rinsed with DPBS and fixed in 80% EtOH at 4 °C for 1 h. After that, the cells were stained with alizarin red S solution (DUKSAN, Seoul, Republic of Korea) at room temperature for 3 min. Lastly, the cells were washed three times and images were captured using a camera. To quantify the alizarin red S staining dye, the dye was extracted using 10% (*v*/*w*) cetylpyridinium chloride (CPC; Sigma-Aldrich, St. Louis, MO, USA) for 15 min. After this, the dye was transferred to new 96-well plates and measured using an ELISA reader at 570 nm. For Von Kossa staining, 1% AgNO_3_ was added and allowed to react under ultraviolet light for 40 min. The plate was then treated with 5% Na_2_S_2_O_3_, rinsed with DW and dried. The plate was monitored with a camera, and the positive area was measured using ImageJ version 1.46 software.

### 4.5. Western Blotting Analysis

In total, 5 × 10^5^ cells in 4 mL medium per dish were seeded in 60π and incubated at 37 °C for 24 h. After this, the cells were treated with PS (25, 50, and 100 μg/mL) for 48 h. Cells were lysed using radioimmunoprecipitation assay (RIPA; composition: 50 mM Tris-Cl, 150 mM NaCl, 1% NP-40, 0.5% sodium deoxycholate, and 0.1% SDS) lysis buffer containing a protease inhibitor cocktail (Sigma, St. Louis, MO, USA). The extracted total proteins were quantified using the BCA protein assay kit and separated by SDS polyacrylamide gel electrophoresis (SDS-PAGE). After this, the proteins were transferred to nitrocellulose membranes using membrane transfer equipment, and the membranes were blocked with 5% skimmed milk for 1 h at room temperature. The membranes were incubated overnight at 4 °C with primary antibodies against RUNX2 (cat. no. ab76956; dilution, 1:1000), BMP-2 (cat. no. ab14933; dilution, 1:1000), p-SMAD1/5 (cat no: 9516S, dilution, 1:1000), SMAD1/5/9 (cat no: ab80255, dilution, 1:500), p-p38 (cat no: 4511S, dilution, 1:1000), t-p38 (cat no: 9212L, dilution, 1:1000), and β-actin (cat. no. sc-8432; dilution, 1:500). Then, the membranes were incubated with secondary antibodies at room temperature for 1 h, and the results were visualized using an enhanced chemiluminescence (ECL) kit (Whatman plc; GE Healthcare), while protein band densitometry was conducted using ImageJ version 1.46 software.

### 4.6. RNA Isolation and RT-PCR Analysis for MC3T3-E1 Cells

In total, 2 × 10^5^ cells in 4 mL medium per dish were seeded in 6-well plates and incubated at 37 °C for 24 h. After this, the cells were treated with PS (25, 50, and 100 μg/mL) for 4 days. Total RNA was extracted using the TRIzol reagent (TaKaRa Bio, Otsu, Japan) according to the manufacturer’s protocols. Total RNA concentration was measured with a Nanodrop 2.0 (Thermo Scientific, Pittsburgh, PA, USA). cDNA was synthesized using a reverse transcription kit (Invitrogen, Carlsbad, CA, USA). RT-PCR was determined using an MG Taq DNA polymerase (MG med, Seoul, Republic of Korea). Amplification was performed using a C1000 Touch™ thermal cycler (Bio-Rad Laboratories; Hercules, CA, USA) with the following cycles: denaturation at 35–40 cycles for 1 min at 94 °C, annealing at 30 s at 55–58 °C, and extension for 1 min at 72 °C. The mouse primers used in this experiment are displayed in Table 1. The PCR products were electrophoresed on 1.2% agarose gel and gene expression was normalized to GAPDH. All the bands were measured using the ImageJ version 1.46 software.

### 4.7. Rat Femoral Fracture Model

Male Sprague Dawley (SD) rats (8 weeks old) were obtained from Raon Biotech, Co., Ltd. (Seoul, Republic of Korea). The animals were housed at 23 ± 2 °C with a 12 h light/dark cycle and free access to food and water. All animal experiments were conducted in accordance with the guidelines for the care and use of laboratory animals and approved by the Committee on Animal Experimentation of Kyung Hee University (Permit Number: KHUASP(SE)-15-095). Twenty-four SD rats were stabilized for one week before being induced with fractures to adapt to the new environment and were divided into three groups: a (i) control group, (ii) PS-low (L) group (a group administered PS dissolved in distilled water at 50 mg/kg after fracture induction), and (iii) PS-high (H) group (a group administered PS dissolved in distilled water at 100 mg/kg after fracture induction). To induce fracture in the model, all rats were anesthetized with 100% oxygen and 5% isoflurane and the right femur skin around the knee was disinfected. The shaved skin was longitudinally incised from the knee to the femoral head. After this, fascia lata and muscle superficials were separated to expose the femur. To induce the fracture, the mid-shaft of the femur was cut using a chainsaw. The fascia lata and skin were stitched with sutures. After waking from the anesthesia, all animals were put back in cages with food and water. Following the operation, we prevented infection at the surgical site with intraperitoneally injected gentamicin at a dose of 4 mg/kg for 3 days. There were no specifics during the operation and all animals were free of wound infection, and none died during surgery. Oral administration of PS was administered once daily at the same time. The administration of PS was conducted for 4 weeks, and body weight was measured once a week. The criteria for humane termination in this experiment were as follows: (1) severe lacerations, bleeding, and infection at the surgical site; (2) when walking to consume food or water becomes too uncomfortable; (3) weight loss of more than 20% compared with the control group of the same age; (4) vomiting and hemoptysis; (5) anxiety and headache. In this study, no animals exhibited abnormal behavior. After end of treatment, all rats were anesthetized with 100% oxygen and 5% isoflurane and sacrificed by lethal cardiac puncture and cervical vertebrae dislocation after the 4-week experimental period, and liver, femur, and blood samples were collected.

### 4.8. Micro-CT and Evaluation of Fracture Recovery Ability

The femurs were analyzed using micro-CT. The micro-CT scanning was conducted using a SkyScan 1176 scanner (SkyScan, Aartselaar, Belgium). Bone microarchitecture parameters such as the bone volume (BV)/total volume (TV) and trabecular thickness (Tb.Th) were assessed using the NRecon software (SkyScan version 1.6.10.1; Bruker Corporation, Billerica, MA, USA). The fracture score was assessed based on three criteria: the degree of callus formation, the callus density, and the stage of fracture line recovery observed in the micro-CT images. Scores were assigned as follows: no recovery = 1, partial recovery = 2, and full recovery = 3.

### 4.9. Whole Blood and Serum Analysis

To extract RNA from whole blood, an equal volume of TRIzol reagent was added to 1 mL of whole blood. Thereafter, cDNA was synthesized and amplified using primers. The rat primers used in this experiment are shown in Table 2. The extracted blood was incubated at room temperature for more than 30 min and then centrifuged at 2000 rpm for 30 min using a centrifuge machine. After that, the supernatant was collected and used for the experiment. ALP, OSN, and OCN, indicators of bone formation in serum, and AST and ALT, indicators of liver toxicity, were verified through ELISA.

### 4.10. Histopathological Examination

Femurs were fixed in 10% neutral buffered formalin (NBF) at room temperature for 24 h and demineralized using ethylenediaminetetraacetic acid (EDTA) for 3 weeks. The samples were then dehydrated with a series of ethanol, xylene, and paraffin. Paraffin blocks were sliced using a rotary microtome (5 μm-thick, ZEISS, Oberkochen, Germany). To determine the morphological changes in the callus and cartilage area, hematoxylin and eosin (H&E) staining was performed. To proceed with IHC staining, femur sections were deparaffinized using xylene and ethanol, and antigen retrieval was performed using proteinase K. Then, following reaction with 0.3% H_2_O_2_ in MetOH for 30 min, the sections were incubated in primary antibody at 4 °C for 24 h (BMP-2 (cat. no. ab14933; dilution, 1:100), RUNX2 (cat. no. ab76956; dilution, 1:50), and anti-OCN (cat no: ab93876, dilution, 1:50)). Thereafter, the secondary antibody was incubated at RT for 1 h and then ABC solution (vector, cat. no: PK6100) at RT for 1 h. After washing three times with TBS, staining with DAB solution (3,3’-diaminobenzidine, vector, cat. no: SK4100) was conducted, then counterstaining with hematoxylin, and the stained tissue was then viewed under a light microscope (Olympus Corporation, Tokyo, Japan).

### 4.11. HPLC Analysis

All quantitative analyses of the PS were performed on a Waters 2695 Alliance System with a 2996 Photodiode Array detector. The samples and standard were assessed on a Waters 2487 Dual λ absorbance detector and X-bridge C18 Column (250 mm × 4.6 mm, 5 μm). The mobile phase was composed of solvent A (acetonitrile) and solvent B (1% acetic acid), and the injection volume of samples and standards was 5 μL. The detection time was 0–30 min, and the flow rate was 1.0 mL/min.

### 4.12. Statistical Analysis

Data were analyzed using GraphPad Prism 5 software (GraphPad Software Inc., La Jolla, CA, USA). Data were presented as the mean ± standard error of mean (SEM). All the experiments were repeated at least 3 times. Significance was examined by one-way analysis of variance (ANOVA) followed by Tukey post hoc analysis. A *p*-value of <0.05 indicated statistical significance.

## Figures and Tables

**Figure 1 ijms-24-07388-f001:**
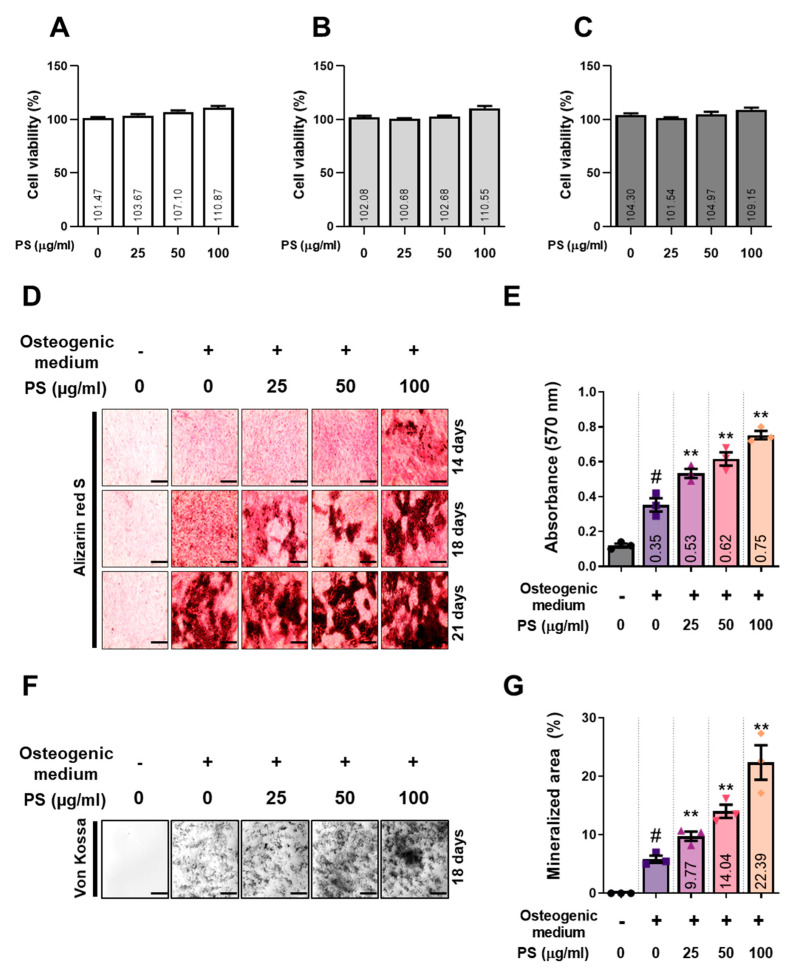
Osteoblast differentiation and bone mineralization are upregulated in PS-treated MC3T3-E1 cells. (**A**–**C**) Cell cytotoxicity assessed via CCK-8 assay for 1, 3, and 7 days. (**D**) Osteoblast differentiation verified by alizarin red S staining (×100, scale bar 200 μm). (**E**) The absorbance of the extracted alizarin red S dye on day 18 measured by ELISA. The bone mineralization area (**F**) stained with Von Kossa stain and (**G**) measured using ImageJ software. All experiments were repeated at least three times. The shapes (triangles, squares, etc.) shown in each graph are values for each round of the experiment. Data are expressed as mean ± SEM. # *p* < 0.05 versus the non-treated cells and ** *p* < 0.01 versus the cells treated only with osteogenic medium.

**Figure 2 ijms-24-07388-f002:**
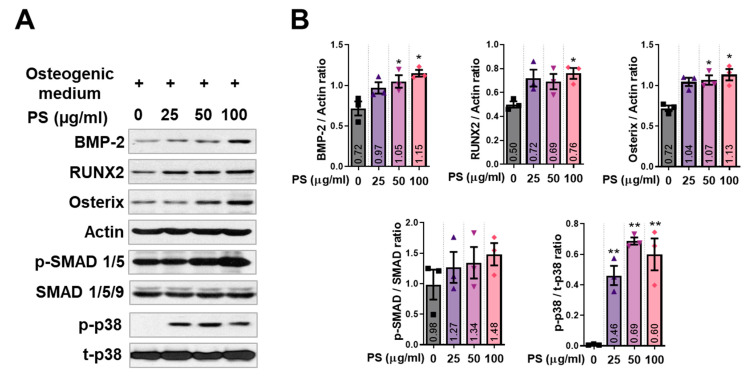
BMP-2 signaling indicators and phosphorylation of p38 were elevated at the protein level in response to PS treatment. (**A**) The effects of PS on BMP, RUNX2 and osterix expression and the phosphorylation of SMAD and p38 were verified by Western blotting. (**B**) Each protein was quantified via comparison with actin, SMAD1/5/9 and t-p38. All experiments were repeated at least three times. The shapes (triangles, squares, etc) shown in each graph are values for each round of the experiment. Data are expressed as mean ± SEM. ** *p* < 0.01, * *p* < 0.05 versus the cells treated only with osteogenic medium.

**Figure 3 ijms-24-07388-f003:**
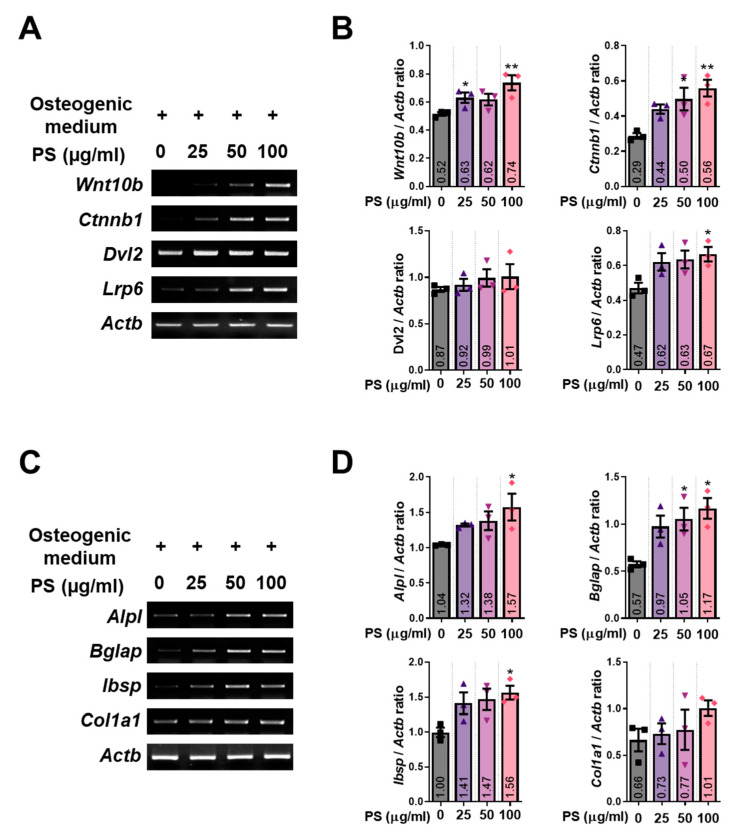
Wnt/β-catenin signaling indicators were elevated at the mRNA level in response to PS treatment. (**A**) The effect of PS on the Wnt/β-catenin mechanism verified by RT-PCR. (**B**) The expression of each gene was quantified through normalization with *Actb*. Osteoblast activation factors were elevated at the gene level due to PS treatment. (**C**) The effect of PS on the expression of osteogenesis-related genes was verified through RT-PCR. (**D**) Expression of each gene was quantified through normalization with *Actb*. All experiments were repeated at least three times. The shapes (triangles, squares, etc) shown in each graph are values for each round of the experiment. Data are expressed as mean ± SEM. ** *p* < 0.01, * *p* < 0.05 versus the cells treated only with osteogenic medium.

**Figure 4 ijms-24-07388-f004:**
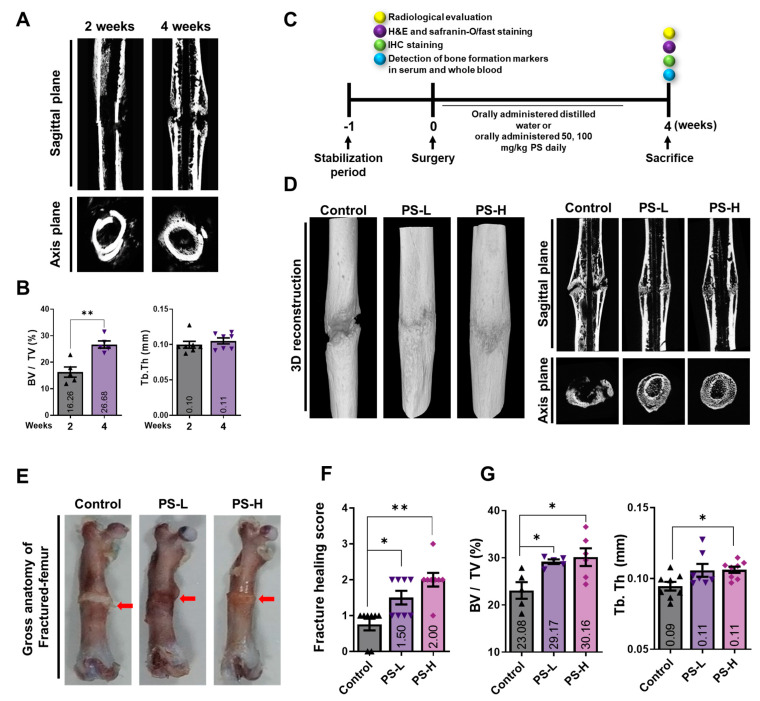
The bone-union-promoting effect of PS verified using a rat model of femoral fracture. (**A**) Preliminary study to set the experimental period. The femur was imaged using micro-CT following sacrifice at 2 and 4 weeks after fracture induction. (**B**) Changes in bone microstructure analyzed using the SkyScan software. (**C**) Design of experiments to evaluate the effects of PS on femoral fracture models. (**D**) Imaging of the fractured femur using 3D reconstruction, sagittal plane, and axis plane methods and micro-CT analysis equipment. (**E**) Gross anatomy of the fractured femur extracted and photographed using a camera for analysis, with the position of the fracture line indicated by the red arrow. (**F**) Healing of fractured femurs scored through micro-CT. (**G**) Measurement of bone microstructure of the imaged callus measured using SkyScan software. The shapes (triangles, squares, etc.) shown in each graph are values for each experimental rat. Data are expressed as mean ± SEM. ** *p* < 0.01, * *p* < 0.05 versus the control group. Control, fracture and treated with vehicle; PS-L, fracture and treated with 50 mg/kg of PS; PS-H, fracture and treated with 100 mg/kg of PS.

**Figure 5 ijms-24-07388-f005:**
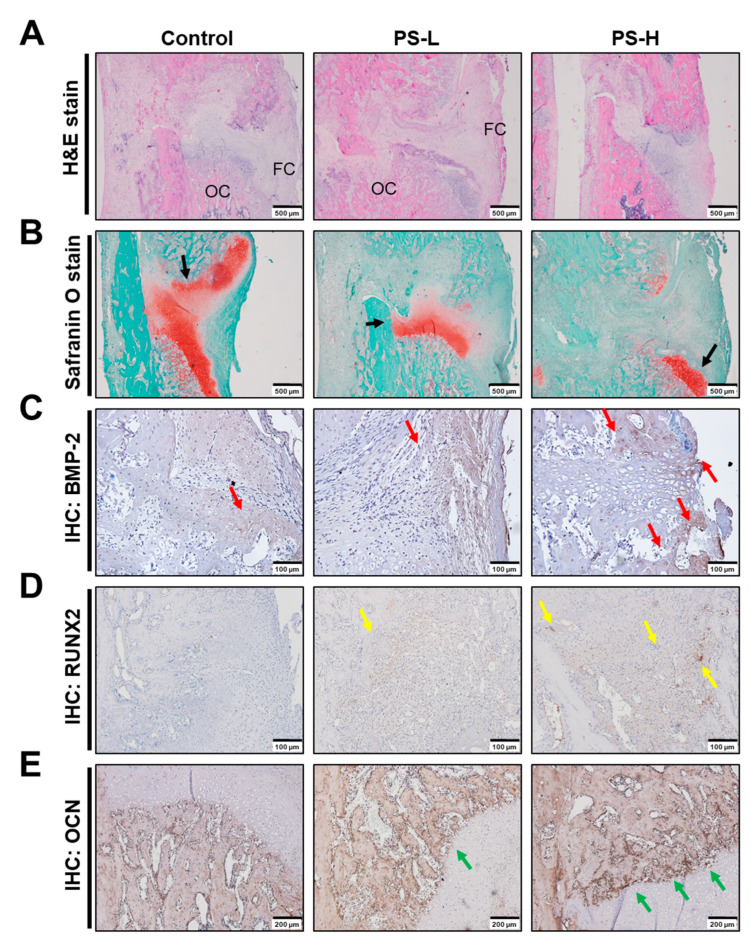
Morphological changes in the fractured femur verified by H&E staining and safranin O staining. (**A**) Changes in fibrous (FC) and osseous calluses (OC) verified by H&E staining (×40, scale bar 500 μm). (**B**) Cartilage areas of fracture sites (black arrow) analyzed by safranin O staining (×40, scale bar: 500 μm). IHC staining was performed to measure the expression of osteogenic factors in femoral tissues. (**C**) BMP-2-positive areas are indicated by red arrows (×200, scale bar: 100 μm). (**D**) RUNX2-positive areas are indicated by yellow arrows (×200, scale bar: 100 μm). (**E**) OCN-positive areas are indicated by green arrows (×100, scale bar: 200 μm).

**Figure 6 ijms-24-07388-f006:**
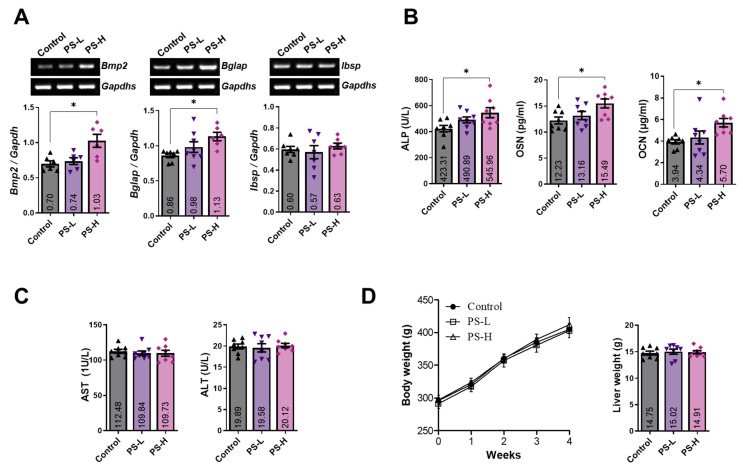
Effects of PS on the expression of osteogenic-related markers in whole blood and serum. Bone formation markers measured (**A**) in whole blood by RT-PCR, with the expression of each marker normalized with GAPDH, and (**B**) in serum by ELISA. (**C**) Hepatotoxicity index, AST/ALT, measured by ELISA. (**D**) Body weight change during weekly PS administration was measured, and the liver weight was measured after sacrifice. The shapes (triangles, squares, etc.) shown in each graph are values for each experimental rat. Data are expressed as mean ± SEM. * *p* < 0.05 versus control group.

**Figure 7 ijms-24-07388-f007:**
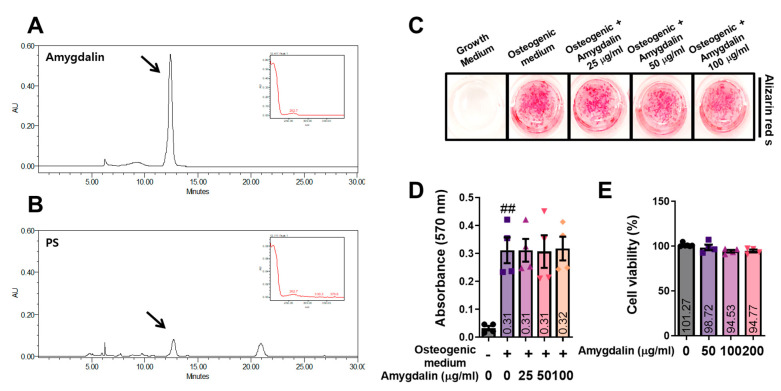
HPLC analysis peaks of (**A**) amygdalin and (**B**) PS, with peak detection at 210 nm. (**C**) The osteoblast differentiation-promoting effect of amygdalin verified by alizarin red S staining. (**D**) Data for absorbance of alizarin red S dye measured by ELISA. (**E**) The cytotoxicity of amygdalin was determined by the CCK-8 assay. The shapes (triangles, squares, etc.) shown in each graph are values for each round of the experiment. ## *p* < 0.01 versus the non-treated cells.

**Figure 8 ijms-24-07388-f008:**
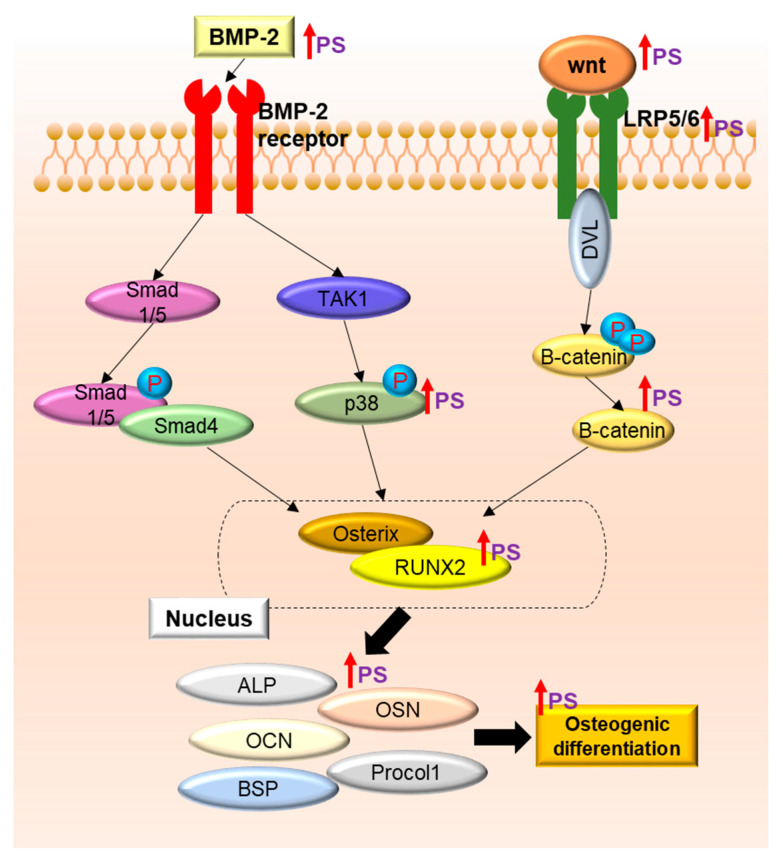
Schematic diagram of PS promotion of osteoblast differentiation (↑: Upregulation function).

**Table 1 ijms-24-07388-t001:** Mouse primer sequences used for RT-PCR.

Source	Gene Name	Sequence (5′-3′)	Accession No.	Tm (°C)	Base Pair
Mouse	*Alpl* (ALP)	F: CGG GAC TGG TAC TCG GAT AA R: TGA GAT CCA GGC CAT CTA GC	NM_001287172.1	55	208
	*Bglap* (OCN)	F: GCA ATA AGG TAG TGA ACA GAC TCC R: GTT TGT AGG CGG TCT TCA AGC	NM_001032298.3	59	147
	*Ibsp* (BSP)	F: AAA GTG AAG GAA AGC GAC GA R: GTT CCT TCT GCA CCT GCT TC	NM_008318.3	53	215
	*Col1a1* (COL1)	F: GCT CCT CTT AGG GGC CAC T R: CCA CGT CTC ACC ATT GGG G	NM_007742.4	60	103
	*Wnt10b* (Wnt10b)	F: TTC TCT CGG GAT TTC TTG GAT TC R: TGC ACT TCC GCT TCA GGT TTT C	NM_011718.2	59	118
	*Ctnnb1* (β-catenin)	F: TGC TGA AGG TGC TGT CTG TC R: CTG CTT AGT CGC TGC ATC TG	NM_001165902.1	59	158
	*Dvl2* (DVL2)	F: GCT TCC ACA TGG CCA TGG GC R: TGG CAC TGC TGG TGA GAG TCA CAG	[51]	64	195
	*Lrp6* (LRP6)	F: CAG CAC CAC AGG CCA CCA A R: TCG AGA CAT TCC TGG AAG AG	[52]	58	220
	*Actb* (β-actin)	F: TTC TAC AAT GAG CTG CGT GT R: CTC ATA GCT CTT CTC CAG GG	NM_007393	58	456

Abbreviations: F, forward; R, reverse; ALP, alkaline phosphatase; OCN, osteocalcin; BSP, bone sialoprotein; COL1, pro-collagen I alpha 1; Wnt10b, formerly Wnt12; β-catenin, beta-catenin; DVL2, dishevelled 2; LRP6, low-density lipoprotein receptor-related protein 6; β-actin, beta-actin.

**Table 2 ijms-24-07388-t002:** Rat primer sequences used for RT-PCR.

Source	Gene Name	Sequence (5′-3′)	Accession No.	Tm (°C)	Base Pair
Rat	*Bmp2* (BMP-2)	F: GAA GCC AGG TGT CTC CAA GAG R: GTG GAT GTC CTT TAC CGT CGT	NM_017178.1	58	122
	*Bglap* (OCN)	F: CAG TAA GGT GGT GAA TAG ACT CCG R: GGT GCC ATA GAT GCG CTT G	NM_013414.1	58	172
	*Ibsp* (BSP)	F: AGA AAG AGC AGC ACG GTT GAG T R: GAC CCT CGT AGC CTT CAT AGC C	NM_012587.2	58	175
	*Gapdhs* (GAPDH)	F: CCT GCA CCA CCA ACT GCT TA R: GGC CAT CCA CAG TCT TCT GAG	NM_017008.4	58	120

Abbreviations: F, forward; R, reverse; BMP-2, bone morphogenetic protein 2; OCN, osteocalcin; BSP, bone sialoprotein; GAPDH, glyceraldehyde 3-phosphate dehydrogenase.

## Data Availability

Not applicable.

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
