# Peer review of "Persicae Semen Promotes Bone Union in Rat Fractures by Stimulating Osteoblastogenesis through BMP-2 and Wnt Signaling"

_ijms, 2023, doi:10.3390/ijms24087388_

Round 1

Reviewer 1 Report

Comments to the authors

  The present manuscript under review titled; “Persicae semen promotes bone union in rat fractures by stimulating osteoblastogenesis through BMP-2/Smad and Wnt/β- catenin signaling”. The paper describes the molecular mechanisms of Persicae semen (PS) concerning the regulation of osteoblast differentiation. The data suggest that PS might be useful for treatment of fracture patient. In general data is appropriately interpreted and analyzed. However, there are several instances where additional discussion would be necessary to support the hypothesis and elevate the impact of the study.

Major points:

1) All in vitro studies have been conducted in the presence of osteogenic medium. Have you tested the effect of PS alone on osteoblast differentiation and signal activation in MC3T3-E1 cells?

2) It is very interesting that that effect of PS is selective on osteoblasts in vivo validation. Please discuss the principal receptor that contributes the biological activity of PS on osteoblasts.

Author Response

Responses to Reviewers:

We would like to thank both of the reviewers for their positive and very helpful comments to improve our manuscript. In response to reviewers, we have addressed the comments as follow:

Reviewer 1.

The present manuscript under review titled; “Persicae semen promotes bone union in rat fractures by stimulating osteoblastogenesis through BMP-2/Smad and Wnt/β- catenin signaling”. The paper describes the molecular mechanisms of Persicae semen (PS) concerning the regulation of osteoblast differentiation. The data suggest that PS might be useful for treatment of fracture patient. In general data is appropriately interpreted and analyzed. However, there are several instances where additional discussion would be necessary to support the hypothesis and elevate the impact of the study.

Major points:

1) All in vitro studies have been conducted in the presence of osteogenic medium. Have you tested the effect of PS alone on osteoblast differentiation and signal activation in MC3T3-E1 cells?

 -> We conducted all of our experiments in osteogenic medium, except for when we were measuring cytotoxicity. While we did not verify the effect of PS alone on the differentiation and mechanism activity of osteoblasts, our goal was to explore its effectiveness in that process. Previous studies have shown that AA and BGP (components of the osteogenic medium) are essential for the differentiation of MC3T3-E1 cells into osteoblasts.

2) It is very interesting that that effect of PS is selective on osteoblasts in vivo validation. Please discuss the principal receptor that contributes the biological activity of PS on osteoblasts.

-> We believe that the primary pharmacological effect is the increase in BMP-2 expression in PS, which is a commonly observed result in both in vitro and in vivo experiments. Therefore, we have discussed the levels of type 1 and type 2 BMP-2 receptors, which are the receptors for BMP-2. Additionally, we have identified through additional experiments the promoting effect of PS on p38. We would like to express our gratitude for the valuable feedback provided by the reviewer (line: 290).

Sentences added in the manuscript: BMP-2 signaling is linked to both type I and type II BMP receptors. When BMP-2 binds to these receptors, it triggers the formation of a complex between the two receptors, which leads to transphosphorylation. This activation of the receptors subsequently activates mechanisms for osteoblast differentiation and bone formation through the use of Smads or MAPKs.

Reviewer 2 Report

This is an interesting study by Jae-Yun Jun et al showing potential beneficial effects of Persicae semen on fracture healing. However, there are still a lot of open questions.

1. Paper needs English editing. The introduction is could be reorganized with some sentences removed (For bone, a fracture refers to a break in continuity due, for example, to strong external impact.).

2. Statements need to be clear and correct. BMP-2 does not phosphorylate SMAD!

3. pSMAD is not increased, so statements throughout the paper need to be corrected and data interpretation modified. Increased BMP2 present in the cultured cells did not lead to a change in pSMAD. To determine the effect on BMP downstream signaling and confirm the effect is active, noncanonical BMP downstream signaling (PI3K, Erk, Jnk) needs to be determined!

4. Line 123. Need to be more precise when talking about activation, which activation, MC3T3-E1 cell activation?

5. How long were cells cultured for evaluation of cell mineralization?

6. Line 137. Therefore, the most important point in the verification of fracture recovery ability is the period from the moment of fracture to the end of the experiment. Clarify sentence.

7. Since the Material and method section is last, you should explain experimental groups when first mentioning them in the text and in the Figure legend.

8. Fig. 5A. Showing only a small part of the callus it is hard to determine the effect of the treatment, especially since it is not mentioned which part of the callus is presented. Staining is weak. Without staining analysis and evaluation of positive signal, with presented figures it is impossible to conclude that there is a difference within the experimental groups. The bone area within the callus and stainings need to be quantified.

9. How was the fracture score evaluated, explain in the Materials and methods.

10. How does the body weight change in this animal study?

11. It is not clear how animals were treated with PS. Explain in the Methods.

12. Data shows that Amigdalyn is not an active ingredient leading to an effect on healing/MC3T3-E3 cells. What are possible compounds of PS that could potentially lead to the healing phenotype?

13. Manuscript could have better mind flow if the Authors would consider re-organizing it. 

14. How long after the fracture were rats sacrificed 3 or 4 weeks? Be consistent.

15. It is not clear how were the rats treated with PS. A diagram of the experimental design would be nice to show.

16. Explain all the abbreviations when first time using them. 

17. Discussion is written as a book chapter and does not discuss much on presented and publish data relevant for the study.

Author Response

Responses to Reviewers:

We would like to thank both of the reviewers for their positive and very helpful comments to improve our manuscript. In response to reviewers, we have addressed the comments as follow:

Reviewer 2.

This is an interesting study by Jae-Yun Jun et al showing potential beneficial effects of Persicae semen on fracture healing. However, there are still a lot of open questions.

  1. Paper needs English editing. The introduction is could be reorganized with some sentences removed (For bone, a fracture refers to a break in continuity due, for example, to strong external impact.).

-> We corrected that sentence in the introduction. In addition, it was reviewed once again to see if there were any incorrectly written sentences in the thesis. The manuscript was proofread on the manuscript proofing sheet designated by the journal prior to submission. We will send you the corresponding certificate along with the answer (line: 34, 228, etc).

  1. Statements need to be clear and correct. BMP-2 does not phosphorylate SMAD!

-> We confirmed that the interpretation of the results was wrong, and after correcting the contents, we revised the overall interpretation of the paper. Sorry for the confusion.

  1. pSMAD is not increased, so statements throughout the paper need to be corrected and data interpretation modified. Increased BMP2 present in the cultured cells did not lead to a change in pSMAD. To determine the effect on BMP downstream signaling and confirm the effect is active, noncanonical BMP downstream signaling (PI3K, Erk, Jnk) needs to be determined!

-> After confirming that pSMAD does not upregulate BMP-2, we went on to verify whether p38 stimulates RUNX2 through the BMP-2 mechanism. P38 was significantly increased by PS stimulation, and the corresponding results are added to figure 2A. We have revised all experimental details, results, and considerations. Thank you to the reviewer for their helpful feedback (line: 110).

Sentences added in the manuscript: In this study, it was observed that PS upregulated the expression of BMP-2 and RUNX2, but only induced a slight increase in the expression of SMAD1/5, which was not statisti-cally significant. Thus, it was concluded that PS upregulated the expression of RUNX2 through other pathways involving Smad. The expression of p38 is crucial for both osteogenesis and BMP-2 expression. Previous studies have shown that inhibiting p38 in osteoblasts completely prevents the gene expression and mineralization of BMP-2 and osteocalcin. Additionally, the p38 inhibitor SB203580 impairs osteoblast differentiation and the expression of osteoblast markers like ALP and OCN in MC3T3-E1 cells. This phenotype is similar to that of RUNX2 mutants, suggesting that p38 is essential for RUNX2's function.

  1. Line 123. Need to be more precise when talking about activation, which activation, MC3T3-E1 cell activation?

-> We recognized the confusion in the sentence and corrected the paper. The corrected sentence is:

Corrected sentences in the manuscript: Activated RUNX2 upregulates the expression of osteoblast-related genes such as ALP, OCN, BSP and procol1 (line: 123)..

  1. How long were cells cultured for evaluation of cell mineralization?

-> Cell mineralization was observed from the 14th and was observed to the extent that the plate was full on the 21st. We added the corresponding content to the Materials and Methods (4.4.) part.

Sentences added in the manuscript: Calcified nodules were observed from the 14th day of culture and continuously increased until the 21st day, and the plate was stained on the 14th, 18th, and 21st days, respectively (line: 442).

  1. Line 137. Therefore, the most important point in the verification of fracture recovery ability is the period from the moment of fracture to the end of the experiment. Clarify sentence.

-> We have corrected that sentence.

Corrected sentences in the manuscript: To determine the extent of recovery through drug administration, it is crucial to establish a suitable endpoint for the experiment after the fracture. If sacrifice occurs too early, callus formation will be incomplete due to inflammatory stages, and if it occurs too late, all fractures in the control group will have undergone recovery. Therefore, setting an appropriate termination time is of utmost importance (line: 138).

  1. Since the Material and method section is last, you should explain experimental groups when first mentioning them in the text and in the Figure legend.

-> We added explanations for each group in the results and figure legend respectively (line: 151, 174).

  1. Fig. 5A. Showing only a small part of the callus it is hard to determine the effect of the treatment, especially since it is not mentioned which part of the callus is presented. Staining is weak. Without staining analysis and evaluation of positive signal, with presented figures it is impossible to conclude that there is a difference within the experimental groups. The bone area within the callus and stainings need to be quantified.

-> We swapped out Figure 5A with a photo taken at a 40x magnification instead of 200x. This change allowed us to confirm more clearly that the fibrous callus area at the fracture site had decreased and that the fracture had healed through PS. Additionally, we re-stained the tissue and took a new picture of Figure 5B at a 40x magnification because the previous staining was too light. We also increased the titer and dab reaction time of IHC RUNX2 and OCN to improve the visibility of the staining. With these modifications, we were able to more clearly observe the fracture recovery ability of PS. Thank you for your positive feedback. However, due to the stained area being spread throughout the callus, it was difficult to measure something specific for quantitative measurement, and therefore, we could not proceed with it. We ask for your understanding (line: 190).

  1. How was the fracture score evaluated, explain in the Materials and methods.

-> Thank you so much for the reviewer's developing comments. We confirmed that the content was insufficiently explained in the manuscript, and added it (line: 523).

Sentences added in the manuscript: The fracture score was assessed based on three criteria: the degree of callus formation, callus density, and the stage of fracture line recovery observed in the micro-CT images. Scores were assigned as follows: no recovery = 1, partial recovery = 2, and full recovery = 3.

  1. How does the body weight change in this animal study?

-> Weight change data of SD rats through fracture and PS administration were added to figure 6. The arrangement of the figure has been modified along with the addition of the corresponding data (line: 214).

  1. It is not clear how animals were treated with PS. Explain in the Methods.

-> We administered the PS extract orally. The contents were additionally described in the materials and methods part of the thesis (line: 497).

Sentences added in the manuscript: Twenty-four SD rats were stabilized for one week before inducing fractures and dividing them into three groups: i) control group, ii) PS-low (L) group (fracture and oral admin-istration of 50 mg/kg of PS), and iii) PS-high (H) group (fracture and oral administration of 100 mg/kg of PS).

  1. Data shows that Amigdalyn is not an active ingredient leading to an effect on healing/MC3T3-E3 cells. What are possible compounds of PS that could potentially lead to the healing phenotype?

-> PS is composed of cyanides, triterpenes, steroids, phenolic acids and fatty acids, and representative components include amygdalin, prunasin, β-sitosterol, campesterol, chlorogenic acid, oleic acid and linoleic acid, among which β-sitosterol, chlorogenic acid and linoleic acid have been shown in previous studies to pro-mote osteoblast differentiation, but a direct experiment using precise PS content and extraction methods would be necessary to confirm this. We acknowledge the limitations of our study and thank the reviewer for their valuable feedback (line: 389).

Sentences added in the manuscript: In this study, we confirmed the osteoblast-activating effect of amygdalin, a major compo-nent of PS. However, this particular component alone did not produce a healing pheno-type. PS is composed of cyanides, triterpenes, steroids, phenolic acids and fatty acids, and representative components include amygdalin, prunasin, β-sitosterol, campesterol, chlorogenic acid, oleic acid and linoleic acid [15,43,44], among which β-sitosterol [45], chlorogenic acid [46] and linoleic acid [47] have been shown in previous studies to pro-mote osteoblast differentiation. If the contents of β-sitosterol, chlorogenic acid and linoleic acid in PS can be precisely analyzed and extracted in future research, it may be possible to identify the specific healing phenotype of PS.

  1. Manuscript could have better mind flow if the Authors would consider re-organizing it. 

-> After adding the results of p38, we reconstructed the manuscript by newly interpreting the osteoblast promotion mechanism of PS. Thank you for your review, and we really appreciate the reviewer's constructive comments.

  1. How long after the fracture were rats sacrificed 3 or 4 weeks? Be consistent.

-> We verified the pharmacological effect of PS by sacrificing 4 weeks after fracture induction. Incorrect entries in the thesis have been corrected. Sorry for the confusion (line 380).

  1. It is not clear how were the rats treated with PS. A diagram of the experimental design would be nice to show.

-> We created an experimental design and added it to figure 4C and reorganized the placement of the figures (line: 168).

  1. Explain all the abbreviations when first time using them. 

-> As the reviewer said, we first explained the abbreviation (line: 76, 106, 151, 162, 163 etc).

  1. Discussion is written as a book chapter and does not discuss much on presented and publish data relevant for the study.

-> We've added further commentary on our experimental findings to aid in their interpretation at the cellular level. Additionally, we've included information on the limitations of the study pertaining to the ingredients. Apart from that, I believe we've adequately addressed everything we intended to. Thank you for reviewing our work.

Round 2

Reviewer 1 Report

The revised manuscript under review titled; “Persicae semen promotes bone union in rat fractures by stimulating osteoblastogenesis through BMP-2/Smad and Wnt/β-catenin signaling”. The authors of the manuscript have adequately answered to all the raised questions. The new additions and modifications have significantly improved the quality of the manuscript.

Author Response

We thank the reviewers for reviewing our manuscript.

Reviewer 2 Report

Jae-Yun et al. improved their manuscript improved some of the statements and edited the English language. However, additional work on the manuscript needs to be done. Some of the statements are still not correct, and English needs to be improved. Some of the sentences that need editing are listed below with other comments that need to be addressed.

1.     A fracture is defined as a state in which the continuity of a bone is completely or incompletely broken due to a strong external impact. Is a definition of a fracture needed? Even a nonscientist knows what the fracture is. If you still decide to keep the definition, I would suggest making clear, correct sentences. There are bones that could be fractured even without external impact. Bones do not break just because of a strong external impact.

2.     Line 36. There is still room for English language improvement. Fractures cause extreme pain to a patient and impair movement, thereby significantly reducing 37 their quality of life.

3.      Line 84. Line 84. Peaches are rich in organic acids and high in sugar content. They are mainly consumed as fresh agricultural products in Korea, and the seeds are traditionally used as a treatment to accelerate blood flow; they have also been used for treating conditions such as neuralgia, muscle pain, menstrual pain, and constipation [11]. The first part of a sentence presumably has nothing with the other part.

4.     Line 337. …the weight of the liver also did not cause a significant difference (Fig. 6D).

5.     Would you expect heterotopic bone formation with prolonged PS treatment? How does PS affect intact bones?

6.     Line 394. The cytotoxicity of amygdalin was verified by CCK-8 assay. Seems like Amygdalin is not toxic, so cytotoxicity determination, not verification.

7.     Line 457. Thus, it was concluded that PS upregulated the expression of RUNX2 through other pathways than Smad.

8.     Discussion is improved, but mechanistic aspects and relevant mechanisms of PS and its components known from the literature are still not presented. Vascularization of the other tissue is improved, which mechanisms are involved in that process? Can you comment/state some of the known literature?

9.     It is still not clear how the animals were treated with PS. Twenty-four SD rats were stabilized for one week before inducing fractures and dividing them into three groups: i) control group, ii) PS-low (L) group (fracture and oral administration of 50 mg/kg of PS), and iii) PS-high (H) group (fracture and oral administration of 100 mg/kg of PS).

Is it oral gavage? In drinking water? How long the treatment last in the text is not stated and Diagram is not precise to determine the treatment window. What does it mean that the rats were stabilized one week before fracture? Did you insert pins for fracture stabilization, plates, some external stabilization, or no stabilization of a fracture?

Author Response

Responses to Reviewers:

We would like to thank both of the reviewers for their positive and very helpful comments to improve our manuscript. In response to reviewers, we have addressed the comments as follow:

Reviewer 2.

Jae-Yun et al. improved their manuscript improved some of the statements and edited the English language. However, additional work on the manuscript needs to be done. Some of the statements are still not correct, and English needs to be improved. Some of the sentences that need editing are listed below with other comments that need to be addressed.

  1. A fracture is defined as a state in which the continuity of a bone is completely or incompletely broken due to a strong external impact. Is a definition of a fracture needed? Even a nonscientist knows what the fracture is. If you still decide to keep the definition, I would suggest making clear, correct sentences. There are bones that could be fractured even without external impact. Bones do not break just because of a strong external impact.
  • We sympathize with the reviewer's words and have deleted the definition of fracture. In addition, after changing the order of the sentences within the paragraph, a sentence related to the patient's economic problems after a fracture was added (line: 34).
  • The revised introduction seems to emphasize the risk of fracture more than before, really appreciate the reviewer's constructive comments.
  • In addition, the manuscript was reviewed once more for grammatical errors through re-editing at the proofreading center.

  1. Line 36. There is still room for English language improvement. Fractures cause extreme pain to a patient and impair movement, thereby significantly reducing 37 their quality of life.
  • We have corrected that sentence (line: 39).

  1. Line 84. Line 84. Peaches are rich in organic acids and high in sugar content. They are mainly consumed as fresh agricultural products in Korea, and the seeds are traditionally used as a treatment to accelerate blood flow; they have also been used for treating conditions such as neuralgia, muscle pain, menstrual pain, and constipation [11]. The first part of a sentence presumably has nothing with the other part.
  • We agree with the reviewer that “Peaches are rich in organic acids and high in sugar content.” sentence has been deleted. It seems like a completely unnecessary sentence in the paragraph. Thank you so much for your nice comments.

  1. Line 337. …the weight of the liver also did not cause a significant difference (Fig. 6D).
  • We have corrected that sentence (line: 212).

  1. Would you expect heterotopic bone formation with prolonged PS treatment? How does PS affect intact bones?
  • In our experimental results, heterotopic bone formation was not observed in the PS group. Referring to the micro-CT results, it can be confirmed that only callus and bone formation for fracture recovery occurred. In addition, since there were no significant abnormalities in the behavior and movement of the mice in the PS-administered group after fracture, it can be assumed that there was no joint pain and movement control, which are the main symptoms of ectopic bone formation. However, we have not been able to conduct detailed verification of intact bone, focusing on the recovery of fracture-induced bone. However, in vitro experiments and fracture-induced models suggest that the bone formation-promoting effect of PS can induce bone formation in intact bone and increase bone strength. Regarding this possibility, we described the limitations of the study #2 at the end of the review (line: 386).

  1. Line 394. The cytotoxicity of amygdalin was verified by CCK-8 assay. Seems like Amygdalin is not toxic, so cytotoxicity determination, not verification.
  • We have corrected that sentence (line: 221, 245).

  1. Line 457.Thus, it was concluded that PS upregulated the expression of RUNX2 through other pathways than Smad.
  • We have corrected that sentence (line: 305).

  1. Discussion is improved, but mechanistic aspects and relevant mechanisms of PS and its components known from the literature are still not presented. Vascularization of the other tissue is improved, which mechanisms are involved in that process? Can you comment/state some of the known literature?
  • We have divided the answer to the content into 2 parts. The first is a mechanistic analysis of PS and its components, and the second is a study of the angiogenic effect and PS. In both parts, the limitations of the study were noted. We really appreciate the reviewer's constructive comments and believe that they greatly improved the quality of the manuscript.
  • First, we compared the osteoblast differentiation-promoting effects of PS and its ingredients to infer which ingredients worked effectively. After that, the contents were written in the last part of the study.
  • Sentences written in the manuscript: among them, β-sitosterol, chlorogenic acid and linoleic acid are known to promote osteoblast differentiation. β-sitosterol upregulates RUNX2 through the induction of ERK and p-38, chlorogenic acid upregulates BMP-2/RUNX2, and finally, linoleic acid upregulates RUNX2 through the Wnt mechanism to promote osteoblast differentiation. Taken together, these results suggest that the main mechanism of the osteoblast differentiation-promoting effect of PS is the upregulation of RUNX2 through various mechanisms of its constituents, and among them, the pharmacological effect of β-sitosterol was found to be similar to that of PS. If the contents of β-sitosterol, chlorogenic acid, and linoleic acid contained in PS can be precisely analyzed and extracted in PS, the specific healing phenotype of PS will be identified (line 395).
  • Second, as we wrote in the Discussion, it has recently been found that angiogenesis is associated with fracture recovery, and based on this, the traditional use method of PS (Hwal-hyeol-yak; 活血藥”, a natural product that speeds up blood circulation) It was referred to when selecting the first drug in anticipation of this efficacy. However, it is not known what effect this effect had through this study. Thus, we listed this in Limitation #4 of the study.
  • Sentences written in the manuscript: PS has demonstrated the ability to effectively restore fractures; however, it remains unclear whether its efficacy is attributed to the traditional method of use, which is believed to enhance blood circulation. Therefore, further research is warranted. To clarify the fracture repair mechanism of PS based on traditional usage, it is recommended to investigate its impact on vascular endothelial growth factor (VEGF) and platelet-derived growth factor (PDGF), which are angiogenic factors associated with fracture repair. Such investigations would allow us to determine the specific effects of PS on these factors and elucidate its potential role in fracture repair (line: 404).

  1. It is still not clear how the animals were treated with PS. Twenty-four SD rats were stabilized for one week before inducing fractures and dividing them into three groups: i) control group, ii) PS-low (L) group (fracture and oral administration of 50 mg/kg of PS), and iii) PS-high (H) group (fracture and oral administration of 100 mg/kg of PS). Is it oral gavage? In drinking water? How long the treatment last in the text is not stated and Diagram is not precise to determine the treatment window. What does it mean that the rats were stabilized one week before fracture? Did you insert pins for fracture stabilization, plates, some external stabilization, or no stabilization of a fracture?
  • We described this in more detail in 4.7 Rat femoral fracture model in Materials and Methods. Stabilization was to adapt the SD-rat received from the company to the new environment of the animal breeding room. Additionally, PS was dissolved in distilled water and injected through the oral cavity (line: 514).

Round 3

Reviewer 2 Report

Dear Authors,

You did modify the manuscript. However, there are still some language errors, with the duration of the PS treatment still not stated. Were both, Low and High treated groups treated with 50mg/kg PS? Please correct the paper.

Best regards.

Author Response

Responses to Reviewers:

We would like to thank both of the reviewers for their positive and very helpful comments to improve our manuscript. In response to reviewers, we have addressed the comments as follow:

Reviewer 2.

Jae-Yun et al. improved their manuscript improved some of the statements and edited the English language. However, additional work on the manuscript needs to be done. Some of the statements are still not correct, and English needs to be improved. Some of the sentences that need editing are listed below with other comments that need to be addressed.

  1. However, there are still some language errors, with the duration of the PS treatment still not stated.
  • We have written about the duration of PS administration. PS was administered for 4 weeks. We apologize for the confusion caused to the reviewer (lines 528, 534).

  1. Were both, Low and High treated groups treated with 50mg/kg PS? Please correct the paper.
  • We corrected the dose of PS-H incorrectly stated in the manuscript (line 518).
